# Parallel CRISPR-Cas9 screens clarify impacts of p53 on screen performance

Anne Ramsay Bowden[†], David A Morales-Juarez[†], Matylda Sczaniecka-Clift, Maria Martin Agudo, Natalia Lukashchuk[‡], John Christopher Thomas*, Stephen P Jackson*

Wellcome/Cancer Research UK Gurdon Institute, University of Cambridge, Cambridge, United Kingdom

**Abstract** CRISPR-Cas9 genome engineering has revolutionised high-throughput functional genomic screens. However, recent work has raised concerns regarding the performance of CRISPR-Cas9 screens using *TP53* wild-type human cells due to a p53-mediated DNA damage response (DDR) limiting the efficiency of generating viable edited cells. To directly assess the impact of cellular p53 status on CRISPR-Cas9 screen performance, we carried out parallel CRISPR-Cas9 screens in wild-type and *TP53* knockout human retinal pigment epithelial cells using a focused dual guide RNA library targeting 852 DDR-associated genes. Our work demonstrates that although functional p53 status negatively affects identification of significantly depleted genes, optimal screen design can nevertheless enable robust screen performance. Through analysis of our own and published screen data, we highlight key factors for successful screens in both wild-type and p53-deficient cells.

**\*For correspondence:**
jct61@cam.ac.uk (JCT);
s.jackson@gurdon.cam.ac.uk (SPJ)

[†]These authors contributed equally to this work

**Present address:** [‡]Oncology Research & Early Development, AstraZeneca, Cambridge, United Kingdom

**Competing interests:** The authors declare that no competing interests exist.

## Introduction

CRISPR-Cas9 genome engineering technologies have transformed cell biology, particularly high throughput functional genomic screens (*Wang et al., 2015*; *Shalem et al., 2014*; *Shalem et al., 2015*; *Smith et al., 2017*) . Pooled CRISPR-Cas9 cell viability screens have been successfully employed in determining gene essentiality (*Hart et al., 2015*), identifying genetic interactions (*Chan et al., 2019*) and assessing drug sensitivities across various genetic backgrounds (*Han et al., 2017*). A number of factors influence CRISPR-Cas9 screen performance, including cellular background. In particular, recent reports concerning technical difficulties in CRISPR-Cas9 genome editing in p53-proficient cells, have brought into question the suitability of p53-proficient cell lines for high throughput CRISPR-Cas9 genetic screens (*Haapaniemi et al., 2018*; *Ihry et al., 2018*).

*TP53,* encoding p53, acts as a master regulator of cell-cycle checkpoint activation (*Kastan et al., 1991*), cellular senescence (*Shay et al., 1991*) and induction of apoptosis in response to DNA damage (*Clarke et al., 1993*; *Lowe et al., 1993*; *Lakin and Jackson, 1999*). *TP53* is arguably the most important tumour suppressor gene, with loss of function mutations in up to 50% of human cancers (*Bouaoun et al., 2016*). Consequently, the p53 status of a cell line, either wild-type (proficient) or mutant (deficient), can be an important factor in determining the suitability of a cellular model, and hence is an important consideration in design of high throughput genetic screens.

Generation of DNA double strand breaks (DSBs) induces p53-dependent cell-cycle arrest in normal fibroblasts (*Di Leonardo et al., 1994*), and most CRISPR-Cas9 genome editing approaches rely on DSB generation to achieve efficient editing (*Jinek et al., 2012*). Recent work has shown that CRISPR-Cas9-associated DSBs in hPSCs (human pluripotent stem cells) induce a p53-mediated apoptotic response, leading to high levels of toxicity and reduced editing efficiency in this background (*Ihry et al., 2018*). Furthermore, a similar p53-mediated DSB response in wild-type retinal pigment epithelial (RPE-1) cells reportedly severely impaired identification of essential genes in a CRISPR-

**eLife digest** The invention of CRISPR-Cas9 genome editing has unlocked a greater understanding of the human genome. Researchers can use this system to make targeted cuts in any gene in the genome, forcing the cell to perform a rapid repair at the cut site. These repairs often introduce mutations into the damaged area, adding or removing DNA letters and disrupting the gene. This allows researchers to study what happens to cells when specific genes are missing, which can help to uncover what each gene is for.

One of the most comprehensive ways to use this technique is to perform a CRISPR-Cas9 screen, which disrupts each gene in the genome one by one. For a CRISPR-Cas9 screen to work well, a cell needs to survive the cuts to its genome. But there is a crucial gene that can stop this happening. Often described as the 'guardian of the genome', this gene codes for a protein called p53, a tumour suppressor that helps to stop a cell turning cancerous when its DNA becomes damaged. This protein activates when the cell senses a cut in its genetic material and can kill the cell if it fails to make a successful repair.

Recent work has shown that the presence of a working copy of the gene for the p53 protein might limit the ability of CRISPR-Cas9 to edit genes. But the evidence was inconclusive. So, Bowden, Morales-Juarez et al. performed two parallel CRISPR-Cas9 screens in human cells with and without p53 to find out more. This revealed that CRISPR-Cas9 can inactivate genes in both normal cells and cells lacking the p53 protein, but that it works better in cells without p53. This was because, when p53 was active, the cells initiated a protective response against the CRISPR-Cas9 cuts. This changed the patterns of genes successfully inactivated by the screen, but it did not make the results unusable. Careful experimental design and thorough data analysis made it possible to get useful results even in cells with functional p53 protein.

The gene for p53 has mutations in around half of human cancers. So, understanding how it affects CRISPR-Cas9 screens could influence the design of future experiments. It is possible that the effects of the p53 protein could vary from cell type to cell type, and with different p53 mutations. Comparisons like the one performed here could help to further unpick how the cell's DNA repair systems might interfere with future CRISPR experiments.

Cas9 screen when compared to RPE-1 *TP53* knockout (*TP53*[KO]) cells (*Haapaniemi et al., 2018*). In contrast, analysis of data from a small number of additional screens in p53 wild-type RPE-1 cells has shown that performance of successful CRISPR screens, as determined by essential gene identification and enrichment of expected targets, is possible in this cellular background (*Brown et al., 2019*). This controversy is confounded by the complexity of variation in experimental design between screens with a lack of controlled parallel experiments. To provide more definitive insights into this important debate, we performed parallel CRISPR-Cas9 screens in paired wild-type and *TP53*[KO] cell lines, thereby minimising additional confounding factors that can preclude accurate screen comparisons.

## Results and discussion

We carried out parallel screens, in wild-type and *TP53*[KO] RPE-1 cells with two independent Cas9-expressing monoclonal populations for each genetic background, selected based on p53 status and high Cas9 cutting efficiency (*Figure 1—figure supplement 1*). To facilitate high screen sensitivity and in-depth interrogation of p53-mediated responses to CRISPR-Cas9-associated DSBs, we designed a bespoke dual guide RNA library targeting 852 DDR-related genes, with 112 olfactory receptor genes included as non-essential gene controls and 14 sequence-scrambled negative controls (*Supplementary file 1*). The library was manually curated to include established DDR components, putatively DDR related interactors, and a considerable number of bioinformatically-associated DDR factors. Moreover, the smaller size of this library compared to a whole genome library enabled high guide representation (>1000 x) to be maintained throughout the screen, minimising the impact of this key factor on screen sensitivity (*Miles et al., 2016*). In addition, our library incorporated a dual guide RNA vector design (*Erard et al., 2017*) to increase the frequency of functional knockout events in transduced cells compared to the canonical single guide RNA (sgRNA) approach. We

reasoned that a vector generating two DSBs per cell may increase detection of differences in screen sensitivity due to variation in DSB responses between genetic backgrounds. Thus, the custom DDR library enables interrogation of p53-mediated DDR events, a cell's overall responses to DSBs, and the fitness effects of inactivating DDR-related genes. Screens were executed as depicted in *Figure 1*, and relative enrichments and depletions of gene knockouts in the edited cell populations were determined from guide read counts generated by next-generation Illumina DNA sequencing (*Supplementary file 2*) using the program MAGeCK (*Li et al., 2014*; *Supplementary file 3*).

In our screens, depletion of core essential genes (as defined by *Hart et al., 2017*) was clearly evident in both wild-type and *TP53^{KO}* backgrounds (*Figure 2A* and *Figure 2—figure supplement 1A*). Due to the conservative nature of this essential gene list, additional genes with significant depletions were also identified in both cell lines (*Supplementary file 3*). A receiver operating characteristic (ROC) curve showing the classification of essential versus non-essential genes by gene depletion p-value ranks (calculated by MAGeCK) (*Figure 2B*) demonstrated good performance of both screens. Nevertheless, the *TP53^{KO}* screen slightly outperformed the wild-type screen at both harvesting timepoints in terms of detection of essential genes by rank.

When the significance of gene depletions was considered, we found that essential genes were much more likely to have low adjusted p-values (q-values) in the *TP53^{KO}* background, compared to wild-type. In addition, we observed that the day 19 timepoint outperformed the day 15 timepoint, detecting increased numbers of essential genes at a given significance threshold (*Figure 2C* and *Figure 2—figure supplement 1B*). The underlying basis behind this differential sensitivity to identifying essential genes lies in the magnitude of the phenotypic effect observed for each guide. While log fold changes (LFCs) across non-core essential ('not essential') genes were not significantly different between the two genetic backgrounds (p=0.60), LFCs for core essential genes were significantly lower in the *TP53^{KO}* screens compared to screens in *TP53* wild-type settings (p=0.0010) (*Figure 2D*), consistent with wild-type cells initiating a p53-mediated response to Cas9-induced DSBs. This would inhibit the proliferation rates of all transduced cells during the course of the screens, leading to smaller LFCs and a narrower distribution of guides within the population, with a consequent reduction in genes with significant depletion scores. Similar results were seen in our analyses of day 15 samples (*Figure 2—figure supplement 1C*).

The impact of the p53-mediated response is also evident when comparing screen results from differential enrichment and depletion of genes between the two genetic backgrounds (*Figure 2E*). As

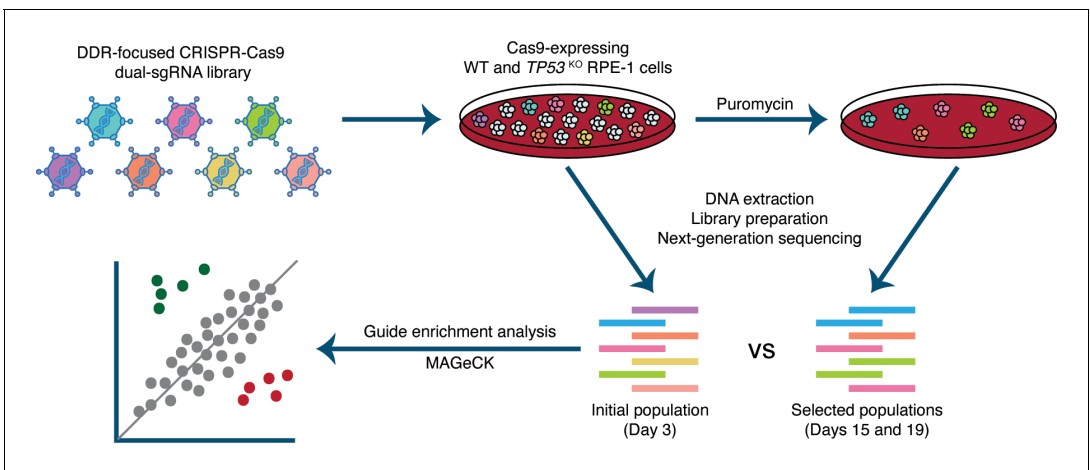

**Figure 1.** Experimental set-up of parallel CRISPR-Cas9 screens in wild-type (WT) and *TP53* knockout(*TP53^{KO}*) RPE-1 cells. Cells were infected at a low multiplicity of infection (MOI=0.3). An initial sample was harvested 48 hours after infection. Subsequently, transduced cells were selected with puromycin and harvested at days 15 and 19. Guide RNA (gRNA) representations were evaluated by extraction of genomic DNA from surviving cells, PCR amplification of barcodes, and next-generation sequencing. MAGeCK (*Li et al., 2014*) was used to determine the relative depletion and enrichment of genes in later samples compared to the 48-hour samples.

The online version of this article includes the following figure supplement(s) for figure 1:

**Figure supplement 1.** Validation of RPE-1 clones used in the screens.

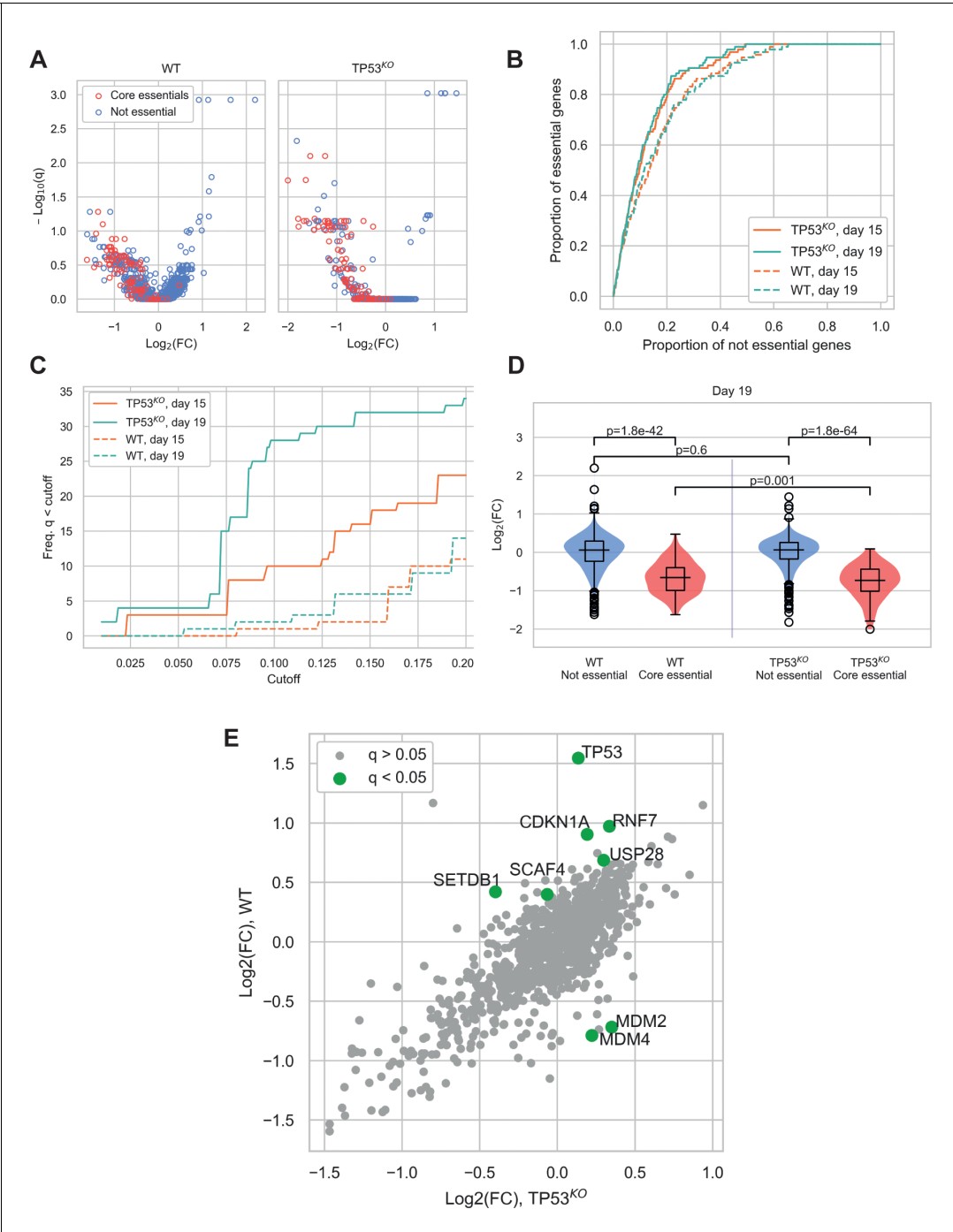

**Figure 2.** Comparison of CRISPR-Cas9 screens in wild-type (WT) and *TP53* knockout(*TP53*^KO) RPE-1 cells demonstrates the impact of p53 on screen performance. (**A**) Mean log$_2$ fold change (LFC) in guide abundance per gene, and significance of this change, from day 3 to day 19 of the experiment. The q-values are false discovery rates (FDR) given by MAGeCK. (**B**) Receiver operating characteristic curves of MAGeCK p-values, discriminating between genes classified as core essential by *Hart et al. (2017)* and other genes. (**C**) Number of core essential genes with q-value less than the range of values given on the x-axis. (**D**) Mean LFC of guides targeting core essential and not core essential genes (Day 19 samples). Paired t-tests were used to test core essential or not essential genes between cell lines, unpaired t-tests were used within a cell line. (**E**) Mean LFC of guides targeting core essential and not core essential genes (Day 19 samples).

The online version of this article includes the following figure supplement(s) for figure 2:

**Figure supplement 1.** Additional comparisons between wild-type and *TP53*^KO CRISPR-Cas9 screens.

**Figure supplement 2.** Biological pathway analysis identifies cell-cycle and p53 signalling as the pathways showing enrichment in the wild-type (WT) compared to *TP53*^KO screens.

expected, in *TP53* wild-type cells, guides targeting *TP53* were the most significantly enriched, with guides targeting other components of the p53 pathway showing the most significant differences between the two genetic backgrounds. Guides significantly enriched in the *TP53* wild-type background included those targeting *CDKN1A* that encodes p21, the major downstream mediator of p53-mediated cell cycle arrest (*el-Deiry et al., 1993*), and those targeting *USP28* that encodes a deubiquitylating enzyme that acts to stabilise p53 (*Zhang et al., 2006*; *Cuella-Martin et al., 2016*). In contrast, guides targeting genes that were significantly depleted in the wild-type but not the *TP53* knockout background included *MDM2* and *MDM4*, which act as negative regulators of p53. MDM2 is an E3 ubiquitin ligase that targets p53 for degradation (*Haupt et al., 1997*), while MDM4 inhibits p53-dependent transcriptional activity (*Francoz et al., 2006*). *SETDB1*, which acts via MDM2, was also enriched in the *TP53* wild-type background. This protein forms a complex with p53 and catalyses p53 K370 di-methylation. Attenuation of SETDB1 reduces the level of di-methylation at this site, leading to increased recognition and degradation of p53 by MDM2 (*Fei et al., 2015*). Furthermore, when we assessed the enrichment/depletion of specific biological pathways between the wild-type and *TP53^{KO}* backgrounds, cell cycle and p53 signalling were the two pathways that were enriched (*Figure 2—figure supplement 2* and *Supplementary file 4*).

Genes that are not acting in the p53 pathway were also identified as significantly enriched (e.g. *EP300*) or depleted (*e.g. CCNA2*) at a FDR < 0.1 (*Supplementary file 3*). *EP300* was enriched on both genetic backgrounds and has an established role as a tumour suppressor through the regulation of the $G_1/S$ cell-cycle transition (*Ait-Si-Ali et al., 2000*). *CCNA2*, or cyclin A2, was depleted on both genetic backgrounds as it interacts with both *CDK1* and *CDK2* to drive S-phase progression and regulate the $G_1/S$ and $G_2/M$ phases of the cell-cycle (*Pagano et al., 1992*). Altogether, these results demonstrate that despite reduced screen sensitivity in p53-proficient cells, biologically meaningful enrichment and depletion analyses at the individual gene and pathway levels can, when required, still be performed in *TP53* wild-type settings.

To further contextualise the feasibility of performing CRISPR-Cas9 screens in a p53-proficient background, we analysed our screens with five others performed in *TP53* wild-type RPE-1 cells. When we performed a comparative ROC curve analysis to assess the screens' abilities to discriminate between core essential genes and other genes (*Figure 3A*), this established that the performance of all screens was similar, with the exception of *Haapaniemi et al. (2018)* data which underperformed in the ability to distinguish essential genes. We then examined the distribution of normalised LFCs for each screen (*Figure 3B*). This revealed that the core essential genes formed distributions distinct from those of olfactory receptors and other non-essential genes in all wild-type screens, with the exception of the Haapaniemi et al. screen where the separation was minimal (the smaller median LFC in our screen compared to the other four successful screens did not notably hinder our ability to distinguish essential genes). Taken together, these analyses provide further evidence that CRISPR-Cas9 screens can be performed successfully in a p53-proficient background. It appears that the Haapaniemi et al. screen is an outlier in its inability to robustly detect essential genes, possibly due to differences in experimental design and execution, and perhaps reflecting relatively low editing efficiency of the polyclonal RPE-1 population used in this screen. This factor strengthens the importance of carefully selecting clones with high Cas9 editing efficiency and also for the use of biological replicates, to enable recognition of common screen results that are independent of clonal background.

We noted that while the median LFC is higher in the LTRI/MDACC, Hart, UBC and MSKCC screens, the variance is also increased when compared to ours. Consequently, we interrogated the relationship between the standard deviation (SD) of the LFCs and the mean LFC values for each of the wild-type screens. *Figure 3C* shows that the variance in LFC between guides targeting the same gene is less in our screen than in these other screens. We speculate that this decrease in variance is linked to the much higher gRNA representation kept throughout our screen (>1000 x mean gRNA representation) than in these other screens, although we cannot discard the possibility that the dual-sgRNA system we used is the cause of this effect. High gRNA representation is relevant for the success and reliability of CRISPR-Cas9 screens, with most published recommendations suggesting screening to at least 200x gRNA representation (*Aregger et al., 2019*) but ideally >500 x (*Joung et al., 2017*). Importantly, high representation must be maintained throughout cell culture and also in the PCR amplification steps. Sufficient sequencing depth is also essential to maintain the sensitivity achieved through high gRNA representation. *Figure 3D* demonstrates the variability in guide abundance determined by sequencing reads across the screens analysed. The MSKCC screen

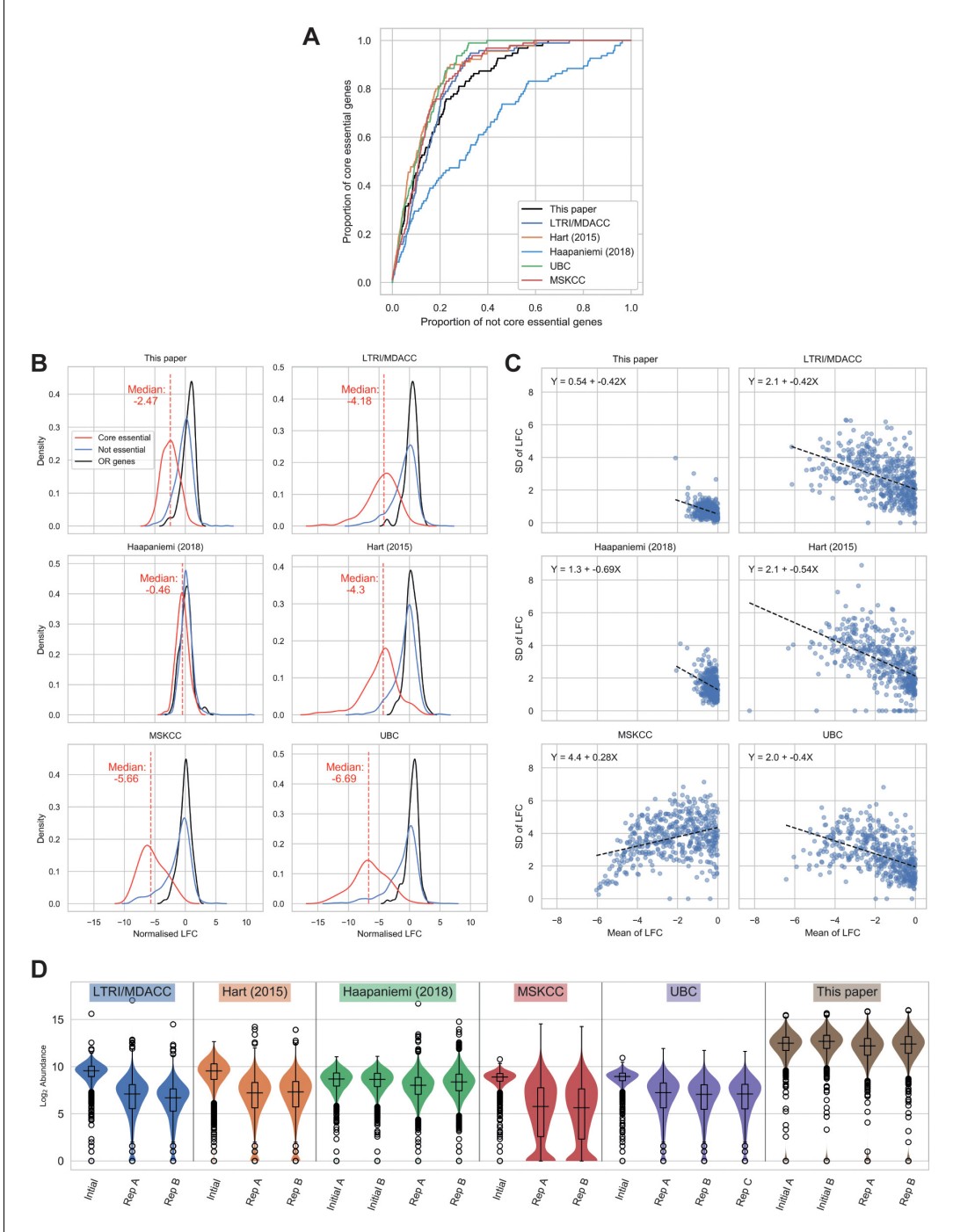

**Figure 3.** Comparison of wild-type (WT) RPE-1 CRISPR-Cas9 screens highlights important factors in screen design. (**A**) Receiver operating characteristic curves of MAGeCK p-values, discriminating between core essential and not core essential genes in *TP53* WT cells. (**B**) Distribution of normalised log$_2$ fold changes (LFCs). The solid lines give kernel density estimates for each distribution, and the dashed line shows the median LFC of the core essential genes. (**C**) Mean LFC vs standard deviation (SD) per gene for genes with mean LFC < 0. As the SD is expected to scale with mean LFC, and the LFC distributions vary between experiments, ordinary least squares regressions were performed to determine the size of the variance across the range of LFCs. The dashed line shows the line of best fit and the equation for each line is given in the chart. (**D**) Log$_2$ guide abundance across all screens. Box plots give median and quartile values.

The online version of this article includes the following figure supplement(s) for figure 3:

**Figure supplement 1.** Reduced variance at higher Log Fold Change is attributable to decreased sequencing reads across multiple guides.

**Figure supplement 2.** The effect on detection of core essential genes at different sequencing read depths in our screens.

is the only dataset to show a distribution with a substantial number of zero reads in the final samples, which accounts for the decreased variance at more negative LFCs in this screen (*Figure 3—figure supplement 1*). Through modelling the effect of decreased sequencing depth in our data, we demonstrate that low read counts can notably decrease screen sensitivity (*Figure 3—figure supplement 2*).

## Conclusions

In summary, we present data from parallel screens in *TP53* wild-type and *TP53*<sup>*KO*</sup> RPE-1 cells, which demonstrate that a p53-mediated response does negatively impact the sensitivity of CRISPR-Cas9 screens. The extent of the impact of *TP53* status on CRISPR-Cas9 screens might vary depending on the cell type being studied, including those with loss-of-function mutations in *TP53* without being fully *TP53* null. It remains to be established precisely how and to what extent different *TP53* mutations, including 'hotspot' mutations, might influence CRISPR-Cas9 screen performance. However, we anticipate that most or all cell lines with an intact *TP53* pathway and proper cell-cycle checkpoint activation would likely recapitulate our findings. Other important factors impacting sensitivity include the guide RNA library used, the magnitude of guide effects, adequate gRNA representation and sufficient sequencing depth. Selection of high-editing efficiency Cas9-expressing cells is also highly recommended and use of biological replicates enables identification of clonal variation. Considering these factors in screen design and execution allows successful CRISPR-Cas9 screens to be carried out in both p53-proficient and p53-deficient cells, thereby fostering new biological insights.

# Materials and methods

## Dual-sgRNA library design

A custom dual-sgRNA library was designed to target 852 genes related to the DNA damage response, 112 olfactory-receptor genes, and 14 sequence scrambled negative controls with a total of 3404 dual-sgRNAs. The genes targeted by this library include a total of 95 core essential genes. The sgRNA sequences and pairwise scores were determined using the Croatan scoring algorithm (*Erard et al., 2017*). Transomic Technologies selected the top pairs of sgRNAs for each gene and assigned a distinct barcode to each pair, cloned them into the pCLIP-*dual*-SFFV-ZsGreen vector, and packaged them into lentiviral particles ready for transduction. For pooled screening, the viral titre was determined by exposing cells to a 6-point dose response of the lentiviral stock. The optimal concentration of virus to achieve a multiplicity of infection (MOI) of 0.3 was determined by linear regression analysis.

## CRISPR-Cas9 screens

CRISPR-Cas9 screens were performed using the custom dual-sgRNA DNA damage response library outlined above. Biological duplicates (two independently isolated Cas9-expressing clones) of wild-type and *TP53*<sup>*KO*</sup> RPE-1 cells were transduced at a MOI of 0.3 and >1,000 fold coverage of the library. The following day, cells were cultured with puromycin to select for the transductants for 12 additional days. Surviving cells from each biological replicate were harvested prior to puromycin selection (day 3), and at day 15 and day 19 after initial transduction. Subsequently, the genomic DNA (gDNA) was isolated using TAIL buffer (17 mM Tris pH 7.5, 17 mM EDTA, 170 mM NaCl, 0.85% SDS, and 1 mg/mL Proteinase K) and subjected to 24 PCR reactions with custom indexed primers designed to amplify the barcode within the lentiviral backbone and append Illumina adapter sequences. Finally, the PCR products were purified (QIAquick PCR Purification kit, Qiagen), multiplexed, and sequenced on an Illumina HiSeq1500 system. Genes enriched or depleted in the day 15 and day 19 samples compared to the day 3 samples were determined using MAGeCK v0.5.9.2 (*Li et al., 2014*).

## Cell culture

RPE-1 *TP53* wild-type and *TP53*<sup>*KO*</sup> cells were cultured in DMEM/F-12 media (Dulbecco's Modified Eagle Medium: Nutrient Mixture Ham's F-12, Sigma-Aldrich) supplemented with 17 mL of 7.5% $NaHCO_3$ (Sigma-Aldrich) per 500 mL, 10% (v/v) foetal bovine serum (FBS, BioSera), 100 U/mL penicillin, 100 μg/mL streptomycin (Sigma-Aldrich), 2 mM L-glutamine, and 10 μg/mL blasticidin (Sigma-

Aldrich) to select for Cas9 expressing cells. Cells were additionally cultured with 1.5 μg/mL puromycin during selection of the transductants.

## Western blot

RPE-1 *TP53* wild-type and *TP53*$^{KO}$ cells were harvested in 100–200 uL of Laemmli buffer (120 mM Tris 6.8 pH, 4%SDS, 20% glycerol). Protein concentrations were determined using a NanoDrop spectrophotometer (Thermo Scientific) at A280 nm. SDS-PAGE was performed with 35 μg of protein lysates, the proteins were resolved on a precast NuPAGE Novex 4–12% Bis/Tris gradient gel (Invitrogen). Resolved proteins were transferred to a nitrocellulose membrane (GE Healthcare) and immunoblotted with the following antibodies at a 1/1,000 dilution: p53 (#554293, BD Biosciences) and GAPDH (#MAB374, Merck Millipore).

## Human cell line generation

RPE-1 wild-type cells were originally obtained from the ATCC cell repository by Professor Jonathon Pines. They were routinely tested for mycoplasma and were authenticated using Affymetrix SNP6 copy number analysis. RPE-1 *TP53*$^{KO}$ cells were generated as described previously (*Chiang et al., 2016*). The *TP53* wild-type and *TP53*$^{KO}$ RPE-1 cells were transduced with a lentiviral vector encoding Cas9 and a blasticidin resistance cassette to facilitate the isolation of Cas9-expressing clones. Limiting dilution of the transduced population enabled isolation of monoclonal cell lines. Cas9 expression was validated by western blot and Cas9 editing efficiency was assayed by transducing clones with a lentiviral vector encoding GFP, BFP, and a sgRNA for GFP (obtained from Dr Emmanouil Metzakopian, UK Dementia Research Institute, Cambridge, UK). Transduced and non-transduced cells were subjected to FACS sorting using an LSRFortessa (BD Biosciences) flow cytometer. The Cas9 editing efficiency for each clone was calculated by comparing the percentage of BFP$^+$ (i.e. edited) cells to the GFP/BFP$^+$ cells (i.e. total transduced population) using FlowJo.

## Statistical software used

Statistical analyses were performed in Python (3.7.5), using the following packages in particular:

- MAGeCK (0.5.9.2)
- jupyterlab (1.1.4)
- matplotlib (3.1.1)
- seaborn (0.9.0)
- pandas (0.25.0)
- numpy (1.16.4)
- scipy (for t-tests & Fisher's exact test, 1.3.0)
- scikit-learn (for PCA, 0.21.2)
- statsmodels (for linear regression and multiple testing correction, 0.10.1)

## CRISPR screen re-analyses

Data files containing guide abundances were downloaded from https://www.ncbi.nlm.nih.gov/geo/query/acc.cgi?acc=GSE128210.

*Supplementary file 5* lists the origins of the data. Where multiple timepoints were available, the day 18 timepoint was used. Guides targeting genes not present in our DDR library were removed from the abundance tables, and MAGeCK (0.5.9.2) was used to obtain significance values for depletion and enrichment of genes. The command line arguments `remove-zero-threshold=10` and `remove-zero=control` were used.

## LFC normalisation

LFCs were normalised by subtracting the mean of the olfactory receptor (OR) genes from all values, and then dividing all values by the SD of the OR genes.

## Resampling

To simulate smaller sequencing runs, guide abundances were resampled by N random draws using the initial abundances as weights. N was set to yield expected median abundances ranging between

10 and 1000. MAGeCK was used to obtain significance values as above. five replicate draws were performed per sample.

## Pathway analysis

Genes within the library were annotated according to KEGG (Kyoto Encyclopedia of Genes and Genomes) pathway. Selection of relevant pathways within the library was based on classifications by *Pearl et al. (2015)*. The enrichment of genes with $p < 0.05$ in these pathways was evaluated using Fisher's exact test. Genes that were depleted over time, or enriched, were tested separately.

## Acknowledgements

We thank all Steve Jackson lab members for support and advice, particularly K Dry who provided editorial assistance with the manuscript. The GFP-BFP-sgRNA vector used for determining Cas9-editing efficiency was provided by Dr Emmanouil Metzakopian (UK Dementia Research Institute, Cambridge, UK).

Research in the SPJ laboratory is funded by Cancer Research UK (programme grant C6/A18796) and Wellcome Investigator Award (206388/Z/17/Z). Institute core infrastructure funding is provided by Cancer Research UK (C6946/A24843) and Wellcome (WT203144). SPJ receives salary from the University of Cambridge. This work was funded by Cancer Research UK programme grants C6/A18796 and C6/A11224 (MS-C and NL, respectively), Wellcome Investigator Award 206388/Z/17/Z (JT and MMA). ARB. is funded by a Wellcome Clinical Fellowship. DMJ is funded by a CONACYT-Cambridge scholarship.

## Additional information

### Funding

| Funder | Grant reference number | Author |
|---|---|---|
| Cancer Research UK | C6/A18796 | Anne Ramsay Bowden<br>David A Morales Juarez<br>Matylda Sczaniecka-Clift<br>Maria Martin Agudo<br>John Christopher Thomas<br>Stephen P Jackson |
| Wellcome | 206388/Z/17/Z | Anne Ramsay Bowden<br>David A Morales Juarez<br>Matylda Sczaniecka-Clift<br>Maria Martin Agudo<br>John Christopher Thomas<br>Stephen P Jackson |
| Cancer Research UK | C6946/A24843 | Anne Ramsay Bowden<br>David A Morales Juarez<br>Matylda Sczaniecka-Clift<br>Maria Martin Agudo<br>Natalia Lukashchuk<br>John Christopher Thomas<br>Stephen P Jackson |
| Wellcome | WT203144 | Anne Ramsay Bowden<br>David A Morales Juarez<br>Matylda Sczaniecka-Clift<br>Maria Martin Agudo<br>Natalia Lukashchuk<br>John Christopher Thomas<br>Stephen P Jackson |
| Cancer Research UK | C6/A11224 | Matylda Sczaniecka-Clift<br>Natalia Lukashchuk<br>Stephen P Jackson |
| Consejo Nacional de Ciencia y Tecnología | 304302648 | David A Morales Juarez |
| Wellcome | 205253/Z/16/Z | Anne Ramsay Bowden |

The funders had no role in study design, data collection and interpretation, or the decision to submit the work for publication.

### Author contributions
Anne Ramsay Bowden, Conceptualization, Resources, Investigation, Visualization, Methodology, Writing - original draft, Writing - review and editing; David A Morales-Juarez, Conceptualization, Validation, Investigation, Visualization, Methodology, Writing - original draft, Project administration, Writing - review and editing; Matylda Sczaniecka-Clift, Maria Martin Agudo, Investigation, Contributed substantially to the collection of data; Natalia Lukashchuk, Resources, Investigation; John Christopher Thomas, Conceptualization, Data curation, Software, Formal analysis, Visualization, Methodology, Writing - original draft, Writing - review and editing; Stephen P Jackson, Conceptualization, Resources, Supervision, Funding acquisition, Project administration, Writing - review and editing

### Author ORCIDs
Anne Ramsay Bowden ⓘ https://orcid.org/0000-0003-1138-4452
David A Morales-Juarez ⓘ https://orcid.org/0000-0001-5370-5512
Maria Martin Agudo ⓘ http://orcid.org/0000-0003-3605-9963
John Christopher Thomas ⓘ https://orcid.org/0000-0003-2425-8412
Stephen P Jackson ⓘ https://orcid.org/0000-0001-9317-7937

### Decision letter and Author response
Decision letter https://doi.org/10.7554/eLife.55325.sa1
Author response https://doi.org/10.7554/eLife.55325.sa2

## Additional files

### Supplementary files
• Supplementary file 1. CRISPR library. A comma separated values table (CSV). Rows contain information about CRISPR plasmids used in screens performed for this paper. For each plasmid, columns give the barcode DNA identifying the plasmid, guide name, symbol of the targeted gene, and the two genomic sequences targeted by the guide RNAs expressed by the plasmid.

• Supplementary file 2. Guide Abundances. A tab separated values table (TSV). Numerical values give the number of reads that mapped to barcode sequences given in *Supplementary file 1* after sequencing DNA. Column headers give information about the samples, the first two characters indicate *TP53* wild-type (WT) or knock out (KO), and the last part of the header specifies the number of days after which the sample was harvested.

• Supplementary file 3. MAGeCK Statistics. Results of all analyses performed with MAGeCK. An Excel workbook with results divided by screen. The worksheet names match the screen names used in the paper. Columns in each worksheet give the $\log_2$ fold change ('lfc'), false discovery rate ('fdr'), -log10 FDR ('fdr_log10'), p-value for enrichment ('pos_p') and p-value for depletion ('neg_p').

• Supplementary file 4. Enrichment of KEGG and GO Terms. A CSV that gives results of Fisher's exact test for enrichment of selected GO and KEGG terms. Each row gives the results for a particular term. Genes that are present in the CRISPR library that match terms are listed in the 'intersection' column. Significance statistics for genes enriched or depleted in the screen are presented.

• Supplementary file 5. Data Sources. A TSV that maps the screen names used in this publication to the identity of the original performers of the screen.

• Transparent reporting form

### Data availability
Data files and scripts to produce Figures 2 and 3 are available from Dryad Digital Repository, https://doi.org/10.5061/dryad.2fqz612kr.

The following dataset was generated:

| Author(s) | Year | Dataset title | Dataset URL | Database and Identifier |
|---|---|---|---|---|
| Bowden AR, Morales-Juarez DA, Sczaniecka-Clift M, Agudo MM, Lukashchuk N, Thomas JC, Jackson SP | 2020 | Parallel CRISPR-Cas9 screens clarify impacts of p53 on screen performance | https://doi.org/10.5061/dryad.2fqz612kr | Dryad Digital Repository, 10.5061/dryad.2fqz612kr |

The following previously published dataset was used:

| Author(s) | Year | Dataset title | Dataset URL | Database and Identifier |
|---|---|---|---|---|
| Brown KR, Moffat J | 2019 | CRISPR screens are feasible in TP53 wild-type cells | https://www.ncbi.nlm.nih.gov/geo/query/acc.cgi?acc=GSE128210 | NCBI Gene Expression Omnibus, GSE128210 |

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
