## [Decision Letter]

**Acceptance summary:**

All reviewers have pointed out the importance, relevance and timely nature of the work. The effect of p53 status on the performance of CRISPR-Cas9 screens has indeed been debated and we believe that your study makes a significant contribution in resolving this issue.

**Decision letter after peer review:**

Thank you for submitting your work entitled "Parallel CRISPR-Cas9 screens clarify impacts of p53 on screen performance" for consideration by *eLife*. Your article has been reviewed by three reviewers, one of whom is a member of our Board of Reviewing Editors, and the evaluation has been overseen by a Senior Editor.

All three reviewers have pointed out the importance, relevance and timely nature of the work. The effect of p53 status on the performance of CRISPR-Cas9 screens has indeed been debated and we believe that your study makes a significant contribution in resolving this issue.

They raised, however, several issues that need to be addressed before acceptance, as outlined below:

1) Please address the textual changes proposed by reviewer 1.

2) Indicate which genes -in addition to those highlighted- are substantially enriched or depleted in the WT screen and whether they belong to specific functional groups. Please discuss/speculate on why these genes were selected in this screen.

3) Provide a rationale for choosing DDR as the targeted screen (reviewer 2).

Please find below all the reviewers' comments:

Reviewer #1:

This is an interesting and timely piece of work.

I have two remarks:

1) The screen is performed in one model cell type/system (retinal pigment epithelial cells). Knowing that the type (G1 arrest, senescence, apoptosis,..) and extend of the p53 response can vary dramatically from one cell type to another, one question that emerges is that how can one generalise the findings? The authors may add a statement in the Discussion about this. Similarly the control line is p53 KO. As stated in the Introduction, 50% of human cancers (cell lines) carry *TP53* loss of function mutations (but are not fully KO). It maybe interesting to introduce some of the hotspot mutations in the authors' preferred model system and ask how much this would influence the results.

2) The authors argue that biologically meaningful enrichment and depletion analyses can still be performed in *TP53* WT cells. This is correct if one is interested in p53 biology. It maybe also important to highlight enrichment or depletion of essential genes that do not act in the p53 pathway, when describing this part of the data.

Reviewer #2:

In this brief report, Bowden, Juarez and colleague address an important issue in the genome editing field: the effect of p53 status on the performance of CRISPR-Cas9 screens. Previous studies had reported that activation of the p53 response by the DSBs induced by CRISPR-Cas9 negatively impacted the outcome of genome wide and targeted screens, presumably by causing the elimination of gene-edited cells.

To directly test this hypothesis, the authors performed pooled, dual guides, targeted CRISPR-Cas9 screens in isogenic WT and p53-KO RPE-1 cells using 852 gRNAs targeting DNA damage response genes, 112 control gRNAs targeting olfactory receptor genes, and 14 scrambled gRNAs.

The experiments were performed in two independent clones per genotype, and at two timepoints (15 and 19 days).

The authors convincingly show that the screens can efficiently identify core essential genes in both genotypes, although the screens performed better in the KO clones: lower q-values and greater average gRNA depletion (log2 Fold change) for essential genes. Reassuringly, known p53 regulators score positive for enrichment and depletion selectively in the WT background.

Finally, their screen in WT RPE-1 cells shows good correlation with similar screens performed in the same cell line by other groups, with the significant exception of Haapaniemi et al., whose screen appears to be an outlier.

Collectively, these results are of substantial interest to the scientific community and will guide the optimal design of future screens. The experiments are well performed, the results are clearly described, and their interpretation seems accurate. The relevant scientific literature is cited. The computational approaches used seem appropriate to me.

I only have a few comments:

1) The rationale for choosing DDR for the targeted screen is not entirely clear and should be discussed.

2) The authors should explicitly indicate how many of the genes targeted by the screen are considered "Core essential".

3) In Figure 2E it appears that several other genes in addition to those highlighted are substantially enriched or depleted in the WT screen. Can the authors comment on those as well?

4) The authors interpret the lower depletion of core essential genes detected in the WT background compared to the KO background (Figure 2D) as likely resulting from the general DDR response induced by the gRNAs in the WT cells. This is certainly a plausible explanation, but another trivial possibility is that the KO clones simply have a faster population doubling compared to WT cells, even in the absence of an exogenous source of DNA damage. To exclude this possibility it would be important to directly compare the growth rate of the 4 RPE-1 clones.

Reviewer #3:

In the present manuscripts the authors compare the differences that might arise in defining essential genes when performing CRISPR screens in p53WT vs p53 deficient cells. This has been somewhat confusing due to reports for instance from the Taipale lab claiming that P53 proficient cells were not suitable to conduct such screens. The reasoning behind is that the breaks generated by the CAS9 would activate a P53-dependent toxic response in cells which can mask the results of these studies and reduce the window to identify bona-fide essential genes.

To address this, the authors have generated a novel CRISPR library for DNA damage response genes, based on dual sgRNAs to favour gene deletion events. The study was conducted by comparing essential genes in P53 WT and KO RPE human cells.

The study is well performed, and clearly shows that gene essentiality screens can in fact be conducted in P53 WT cells. The authors show a few examples of genes that are preferentially lost in both backgrounds, which make sense.

In my opinion, the work is well done and the data provided support the main claim of the authors. Namely, that gene essentiality screens can be conducted in P53 WT cells. Since this has been somewhat confusing in the field of screens, I guess it will be clarifying to have it published.

That being said, I somewhat guess that the initial objective of this study was not to enter into this dispute, but rather a more exciting experiment in the look for genes that are selectively essential for P53-KO cells.

Due to the covid crisis, I sincerely feel in no position to ask the authors for more experiments. But perhaps, they could still work a bit more on their existing data and make a better paper if they do a bit more analysis and representations of their data. For instance; they could provide a comprehensive list of all the genes that are selectively lost in P53-KO cells, and discuss/speculate on why this could be. In addition, they could also provide a list of genes that are essential in P53 WT cells, but not in P53-KO ones. This is relevant as many DDR genes are essential in mice, in a manner that is rescued by P53 loss. Dissecting which ones behave in this manner and which ones not, might turn out to be informative.

In summary, I think the data provided is enough to end the discussion on whether gene essentiality screens can be done in P53 WT cells. Before publication, I would suggest the authors work a bit more in the analysis of their data and the presentation, as the same data can make an even nicer paper for their readers.

---

## [Author Response]

Reviewer #1:This is an interesting and timely piece of work.I have two remarks:1) The screen is performed in one model cell type/system (retinal pigment epithelial cells). Knowing that the type (G1 arrest, senescence, apoptosis,..) and extend of the p53 response can vary dramatically from one cell type to another, one question that emerges is that how can one generalise the findings? The authors may add a statement in the Discussion about this. Similarly the control line is p53 KO. As stated in the Introduction, 50% of human cancers (cell lines) carry TP53 loss of function mutations (but are not fully KO). It maybe interesting to introduce some of the hotspot mutations in the authors' preferred model system and ask how much this would influence the results.

In response to these relevant and important comments, we have added the following statements to the conclusion section of our paper:

“The extent of the impact of *TP53* status on CRISPR-Cas9 screens might vary depending on the cell type being studied, including those with loss-of-function mutations in *TP53* without being fully *TP53* null. It remains to be established precisely how and to what extent different TP53 mutations, including “hotspot” mutations, might influence CRISPR screen performance. However, we anticipate that most or all cell lines with an intact *TP53* pathway and proper cell-cycle checkpoint activation would likely recapitulate our findings.”

2) The authors argue that biologically meaningful enrichment and depletion analyses can still be performed in TP53 WT cells. This is correct if one is interested in p53 biology. It maybe also important to highlight enrichment or depletion of essential genes that do not act in the p53 pathway, when describing this part of the data.

While we understand the reasoning for this comment, our conclusions on the negative effects of an active p53-mediated DDR response in CRISPR-Cas9 screens would still hold regardless of whether one’s interest lies in p53 biology. The data presented in this paper (in Figure 2E and Figure 2—figure supplement 2) intend to highlight the key differences amongst the WT and *TP53^KO^*samples. Nevertheless, we are now discussing additional genes that are significantly depleted and enriched that are not acting in the p53 pathway. The revised manuscript now includes the following text:

“Genes that are not acting in the p53 pathway were also identified as significantly enriched (e.g. *EP300*) or depleted (*e.g. CCNA2*) at a FDR<0.1 (Supplementary file 5). *EP300* was enriched on both genetic backgrounds and has an established role as a tumour suppressor through the regulation of the G_1_/S cell-cycle transition (Ait-Si-Ali et al., 2000). *CCNA2*, or cyclin A2, was depleted on both genetic backgrounds as it interacts with both *CDK1* and *CDK2* to drive S-phase progression and regulate the G_1_/S and G_2_/M phases of the cell-cycle (Pagano et al., 1992).”

Additionally, we are including a table of the genes significantly (FDR<0.1) enriched and depleted for both genetic backgrounds as Supplementary file 5.

Indicate which genes – in addition to those highlighted – are substantially enriched or depleted in the WT screen and whether they belong to specific functional groups. Please discuss/speculate on why these genes were selected in this screen.

In this paper, we were interested in analysing the differences between the WT and *TP53^KO^*samples (as highlighted in Figure 2E and Figure 2—figure supplement 2) but, as the reviewers highlight, it might be interesting to look into functional clustering of the individual samples. Therefore, we performed a functional clustering analysis with GO terms of the substantially enriched/depleted genes. There was no significant enrichment for functional groups at FDR < 0.05 and so we have not shown this data. All the results indicating enrichments and depletions from the individual samples are available in supplementary data.

Reviewer #2:[…]I only have a few comments:1) The rationale for choosing DDR for the targeted screen is not entirely clear and should be discussed.

The bespoke custom library allowed us to interrogate aspects of p53 biology and the DDR, the latter being the main focus of our research. Additionally, the size of our focused library compared to a whole genome library facilitated library construction and logistical considerations of performing parallel CRISPR-Cas9 screens with a thorough gRNA representation (1,000x) in our lab setting. To address these points, we include the following text in the Results and Discussion sections of our paper:

“The library was manually curated to include established DDR components, putative DDR-related interactors, and a considerable number of bioinformatically-associated DDR factors.”

“Moreover, the smaller size of this library compared to a whole genome library enabled high guide representation (>1000x) to be maintained throughout the screen, minimising the impact of this key factor on screen sensitivity (Miles, Garippa and Poirier, 2016).”

“Thus, the custom DDR library enables interrogation of p53-mediated DDR events, a cell’s overall responses to DSBs, and the fitness effects of inactivating DDR-related genes.”

2) The authors should explicitly indicate how many of the genes targeted by the screen are considered "Core essential".

There are 95 core essential genes in our DDR library. We have now added this information in the Materials and methods section concerning the dual-sgRNA library design.

3) In Figure 2E it appears that several other genes in addition to those highlighted are substantially enriched or depleted in the WT screen. Can the authors comment on those as well?

The genes highlighted in Figure 2E are those that show significant differences between the two genetic backgrounds (as calculated by MAGeCK). The list of other substantially enriched or depleted genes for both the wild-type and the *TP53^KO^*are provided in supplementary information.

4) The authors interpret the lower depletion of core essential genes detected in the WT background compared to the KO background (Figure 2D) as likely resulting from the general DDR response induced by the gRNAs in the WT cells. This is certainly a plausible explanation, but another trivial possibility is that the KO clones simply have a faster population doubling compared to WT cells, even in the absence of an exogenous source of DNA damage. To exclude this possibility it would be important to directly compare the growth rate of the 4 RPE-1 clones.

We had considered this possibility and experimentally determined that the clones behaved similarly in terms of growth rates. The reviewer correctly points out that the growth rates of the clones in the absence of DNA damage were not presented. In Author response image 1, we provide a graph representing the unperturbed growth rates of all the RPE-1 clones used in our CRISPR-Cas9 screens.

Reviewer #3:[…]That being said, I somewhat guess that the initial objective of this study was not to enter into this dispute, but rather a more exciting experiment in the look for genes that are selectively essential for P53-KO cells.Due to the covid crisis, I sincerely feel in no position to ask the authors for more experiments. But perhaps, they could still work a bit more on their existing data and make a better paper if they do a bit more analysis and representations of their data. For instance; they could provide a comprehensive list of all the genes that are selectively lost in P53-KO cells, and discuss/speculate on why this could be. In addition, they could also provide a list of genes that are essential in P53 WT cells, but not in P53-KO ones. This is relevant as many DDR genes are essential in mice, in a manner that is rescued by P53 loss. Dissecting which ones behave in this manner and which ones not, might turn out to be informative.

Figure 2E shows all the genes that are significantly differentially essential in the two cell backgrounds. Comparing lists of genes that pass some arbitrary threshold in one cell background but not another could possibly confuse the issue, as gene essentiality is not black and white. A small difference in phenotype could result in a significant dropout in one cellular background and a not-quite-significant dropout in the other. For this reason, we only discuss genes where the difference between cell backgrounds is statistically significant.